# Efficient Test-Time Prompt Tuning for Vision-Language Models

## Abstract

Vision-language models have showcased impressive zero-shot classification capabilities when equipped with suitable text prompts. Previous studies have shown the effectiveness of test-time prompt tuning; however, these methods often require per-image prompt adaptation during inference, which is computationally intensive and limits scalability and deployment. To address this issue, we introduce a novel framework: Self-supervised learning for efficient Test-time Prompt Tuning (Self-TPT). The key feature of Self-TPT is its shift to efficient *predefined class adaptation* through self-supervised learning, thereby avoiding the computation-heavy *per-image adaptation* at inference. Self-TPT starts by co-training the self-supervised and supervised tasks using source data, then applies the self-supervision exclusively for new class understanding before making predictions. Specifically, we propose Contrastive Prompt Learning (CPT) as the core task for self-supervision. CPT is designed to minimize the intra-class distances while enhancing inter-class distinguishability via contrastive learning. Empirical evidence suggests that CPT can partially mimic supervised learning in terms of gradients, providing a plausible explanation for its effectiveness. Motivated by this finding, we introduce a gradient matching loss to explicitly enhance gradient similarity. We evaluated Self-TPT across three challenging zero-shot benchmarks. The results consistently show that Self-TPT not only significantly reduces inference costs but also achieves state-of-the-art performance, effectively balancing the efficiency-efficacy trade-off.

## 1 Introduction

Open-set image classification is a fundamental yet challenging task in computer vision. Recently, Vision-Language Models (VLMs) (Jia et al., 2021; Li et al., 2022; Alayrac et al., 2022; Fang et al., 2023) have shown promising capabilities in this field. A prominent model, CLIP (Radford et al., 2021), encodes both images and language into a unified embedding space, allowing classification by measuring similarities between image representations and textual class descriptions.

Effective prompts for input classes are essential (Radford et al., 2021), but manually crafting them is time-consuming (Zhou et al., 2022b). Inspired by NLP advancements (Shin et al., 2020; Zhong et al., 2021), researchers have explored using continuous vectors as soft prompts, optimizing them with a few labeled data (source data). These methods can automatically obtain task-specific prompts, thereby improving performance (Zhou et al., 2022b; Zhu et al., 2024). However, the source data is unlikely to encompass *all* possible classes, resulting in suboptimal *open-set* performance.

Test-time adaptation (TTA) (Liang et al., 2020a; Wang et al., 2020; Niu et al., 2023) has recently gained attention for adapting models to new data distributions during the test phase. In this context, TPT (Shu et al., 2022) was proposed to tune prompts for new classes at test time, thereby improving open-set generalization. As depicted in Fig. 1(a), TPT first learns prompts from source data (`stage 1`). It then adjusts these prompts for each test sample (`stage 2`) and uses the sample-tailored prompts for predicting (`stage 3`). Despite its effectiveness, TPT requires a substantial computational budget. *For each test sample*, it takes multiple forward and backward passes through the entire model and needs to retain the full computational graph, resulting in substantial latency and memory usage. As shown in Fig. 1(c), TPT operates at ∼7 FPS while consuming ∼5GB of graphics memory. The latest TPT-based method, PromptAlign (Samadh et al., 2023), operates at ∼5 FPS

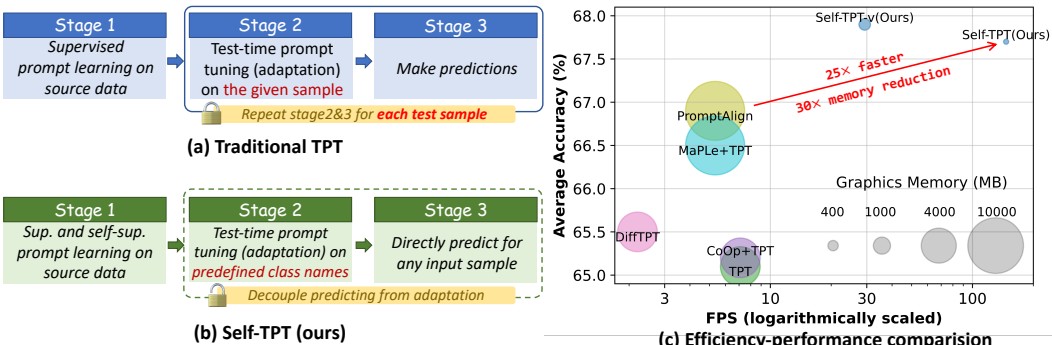

Figure 1: **TPT versus Self-TPT.** (a) TPT learns prompts from source data (`stage 1`), then adapts them to individual samples for prediction (`stages 2&3`). (b) Self-TPT employs text-oriented self-supervised learning (SSL) for joint training (`stage 1`) and for new class adaptation (`stage 2`), followed by *direct* predictions for each image (`stage 3`). (c) We present the frame per second (FPS) and graphics memory usage for each method when applied to CLIP-B/16 using the same A100-80G GPU. The y-axis represents the average cross-dataset accuracy.

with ∼11GB of graphics memory. These heavy computational demands pose challenges for scaling TPT to larger VLMs and deploying it on resource-limited platforms.

In this paper, we study the question: *Can we harness TTA's generalizability while bypassing its huge computational overhead?* We observed that, at inference, while images come sequentially, the candidate class names are typically predetermined. Motivated by this, we introduce Self-TPT, an efficient Test-time Prompt Tuning framework that employs *text-oriented* Self-supervised learning (SSL). The idea is depicted in Fig. 1(b): the adaptation process (`stage 2`) of Self-TPT operates solely on the predefined class names, allowing for *directly* predicting any image without the need for prompt updates (`stage 3`), thereby effectively bypassing the substantial computational load.

For the SSL component, we propose using contrastive learning. The intuition is that effective classification requires embeddings of the same class to align closely, while those from different classes remain distinct, ensuring clear inter-class separability. To achieve this, we introduce a novel approach called Contrastive Prompt Tuning (CPT). In CPT, we vary the insertion points of the class token within prompt sequences as data augmentation to create positive pairs. Negative pairs are built by contrasting a class against all others, thereby explicitly reinforcing the dissimilarity among class embeddings. Initially, we integrate CPT with supervised learning (`stage 1`) and subsequently rely exclusively on CPT for new class adaptation (`stage 2`).

Our empirical analysis shows that CPT and supervised learning consistently exhibit a positive gradient correlation in nearly all cases. This suggests that both tasks drive the model's optimization in similar directions, allowing CPT to potentially serve as a proxy for supervised learning during the adaptation phase. This evidence plausibly explains the effectiveness of CPT. Inspired by this, we introduce a gradient matching (GM) loss designed to enhance this positive correlation explicitly. The GM loss operates on the gradients derived from both supervised and CPT losses and aims to maximize their cosine similarities.

We evaluated Self-TPT's performance on three challenging zero-shot benchmarks: cross-dataset generalization, base-to-new generalization, and domain generalization. Our findings reveal that Self-TPT consistently surpasses the prior state-of-the-art, PromptAlign, improving by 0.93%, 1.59%, and 1.82% on these benchmarks, respectively. Notably, Self-TPT significantly enhances efficiency, as shown in Fig. 1(c), achieving a 25-fold increase in inference speed and reducing memory usage by 30-fold compared to PromptAlign. Our experiments also verify that Self-TPT is data-efficient and generalizes well across different backbones, scales, and VLMs.

## 2 RELATED WORK

**Vision-Language Models.** Recent advances in computer vision and natural language processing have spurred significant interest in vision-language models (VLMs) (Radford et al., 2021; Jia et al.,

2021; Li et al., 2022; Zhai et al., 2022; Alayrac et al., 2022; Fang et al., 2023). These models excel in various multi-modal tasks by leveraging massive datasets of image-text pairs to develop robust representations that span different modalities. Notably, CLIP (Radford et al., 2021) has demonstrated exceptional open-vocabulary capabilities, enabling effective performance in image classification (Zhou et al., 2022b; Zhu et al., 2024), video recognition (Weng et al., 2023; Huang et al., 2024), and object detection (Du et al., 2022; Minderer et al., 2022), *etc.* A key aspect of deploying VLMs successfully involves crafting effective text prompts. In this paper, we introduce a novel framework that optimizes prompt adaptation for better class comprehension during testing.

**Prompt Learning.** Recent advancements in NLP have inspired approaches like CoOp (Zhou et al., 2022b; Chen et al., 2022b; Lu et al., 2022), which treats prompts for VLMs as continuous vectors. However, CoCoOp (Zhou et al., 2022a) highlighted a significant flaw: these learned prompts often overfit to seen classes, compromising performance on new ones. To mitigate this, recent studies have introduced additional learnable components (Zhu et al., 2024; Zhou et al., 2022a; Khattak et al., 2023a; Zang et al., 2022; Wang et al., 2023c; Singha et al., 2023) or specialized strategies (Lee et al., 2023; Xu et al., 2023b; Long et al., 2023; Shi & Yang, 2023; Kan et al., 2023; Wang et al., 2023a) to enhance prompt generalization. Techniques like distillation and regularization (Yao et al., 2023; Zhu et al., 2022; Khattak et al., 2023b; Xu et al., 2023a; Bulat & Tzimiropoulos, 2023) are also employed to integrate task-specific knowledge and hand-crafted priors. Despite progress, achieving prompts that generalize across all possible classes remains challenging. Consequently, this paper shifts focus to test-time adaptation strategies, dynamically adjusting prompts during testing to address open-world application challenges.

**Test-Time Adaptation** was developed to address shifts in data distribution between training and testing phases by dynamically adjusting the model at inference. Numerous methods have emerged, including entropy minimization (Liang et al., 2020a; Wang et al., 2020; Niu et al., 2023), batch-normalization activation (Wang et al., 2020; Zhao et al., 2023; Niu et al., 2023), pseudo labeling (Chen et al., 2022a; Liang et al., 2020b; Wang & Wibisono, 2022), feature alignment (Liu et al., 2021; Wang et al., 2023b; Wang & Aitchison, 2022), and test-time training (Sun et al., 2020; Liu et al., 2021; Huang et al., 2021; Chen et al., 2023; Gandelsman et al., 2022). Recently, test-time prompt tuning (TPT) (Shu et al., 2022) for VLMs has gained attention. TPT optimizes prompts by reducing prediction entropy and uses image augmentation to create prediction diversity. Techniques like DiffTPT (Feng et al., 2023) employ Stable Diffusion (Rombach et al., 2022) to boost augmented image diversity, while SwapPrompt (Ma et al., 2023) and PromptAlign (Samadh et al., 2023) focus on maximizing prediction agreement and aligning token statistics, respectively. Despite their effectiveness, these approaches often entail high computational costs, posing challenges for practical deployment. This study introduces a more efficient TPT framework, aiming to balance effectiveness with real-world applicability.

## 3 METHOD

### 3.1 PRELIMINARIES

**Contrastive Language-Image Pre-training (CLIP)** (Radford et al., 2021) encodes image $x$ and a set of class descriptions $\{t_i\}_{i=1}^{C}$ into a joint vision-language embedding space using two encoders: the image encoder $f(\cdot)$ and the text encoder $g(\cdot)$. Here, $C$ denotes the number of candidate classes. This space ensures conceptually similar inputs are closely aligned. In this way, the classification problem can be formulated as an image-text matching problem. Specifically, CLIP computes the encoded image feature $e$ and the encoded text features $\{w_i\}_{i=1}^{C}$. The probability for the $i$-th class is calculated as:

$$p(y_i \mid x) = \frac{\exp\left(\cos\left(\boldsymbol{w_i}, \boldsymbol{e}\right)/\tau\right)}{\sum_{j=1}^{C} \exp\left(\cos\left(\boldsymbol{w_j}, \boldsymbol{e}\right)/\tau\right)}, \quad (1)$$

where $\tau$ is a temperature parameter.

**Prompt Learning** (Zhou et al., 2022b;a; Yao et al., 2023; Zhu et al., 2022) aims to optimize soft prompts $\mathbf{P} \in \mathbb{R}^{M \times d}$ to replace manually designed prompts (*e.g.*, "a photo of a [CLASS]"):

$$\boldsymbol{t}_i = [P]_1 [P]_2 \dots [P]_M [\texttt{CLASS}]_i. \quad i = 1, \dots, C \quad (2)$$

Here, $[\texttt{CLASS}]_i$ are the word embeddings for the $i$-th class and $d$ is the word embedding dimension of CLIP. The parameters of $\mathbf{P}$, represented as $\boldsymbol{\theta}_p$, are shared across all classes. The training dataset (*i.e.*,

Figure 2: **Overview of Self-TPT.** Self-TPT operates in three stages. `Stage 1`: Conduct prompt learning on a source dataset, co-trained with a self-supervised loss. `Stage 2`: Perform test-time adaptation (TTA) for new class understanding via the self-supervised loss. `Stage 3`: Directly predicting the target dataset *without* further prompt adjustment.

source data), denoted as $\mathcal{S} = \left(\mathcal{X}^{(s)}, \mathcal{Y}^{(s)}, \mathcal{M}^{(s)}\right)$, includes images $\mathcal{X}^{(s)}$, candidate class names $\mathcal{Y}^{(s)}$, and a ground-truth mapping $\mathcal{M}^{(s)}$ between them. Training employs a cross-entropy loss function:

$$\min_{\boldsymbol{\theta_p}} \mathcal{L}_{ce}\left(\mathcal{X}^{(s)}, \mathcal{Y}^{(s)}, \mathcal{M}^{(s)}; \mathbf{P}, f, g\right). \tag{3}$$

During this training phase, the image and text encoders from CLIP are kept frozen.

**Test-time Prompt Tuning (TPT)** (Shu et al., 2022; Feng et al., 2023; Samadh et al., 2023) aims to dynamically adapt prompts to unlabeled test data $\mathcal{T} = \left(\mathcal{X}^{(t)}, \mathcal{Y}^{(t)}\right)$, where $\mathcal{X}^{(t)}, \mathcal{Y}^{(t)}$ represent the target images and candidate class names, respectively. TPT involves three stages: Initially, training prompts on source data. Then, using the prompts from first stage, TPT employs an unsupervised loss, such as entropy minimization $\mathcal{L}_{ent}$, to tailor these prompts for each specific test sample $\mathcal{X}_i^{(t)}$:

$$\min_{\boldsymbol{\theta_p}} \mathcal{L}_{ent}\left(\mathcal{X}_i^{(t)}, \mathcal{Y}^{(t)}; \mathbf{P}, f, g\right). \tag{4}$$

Subsequently, predictions are made for each sample using these tailored prompts. Despite their effectiveness, these methods are computationally intensive during inference. Each test sample requires multiple model passes and retention of a full computational graph, leading to increased latency and significant memory demands. These limitations make deployment challenging, particularly in resource-constrained environments, and hinder scalability to larger VLMs.

## 3.2 PIPELINE OF SELF-TPT

To leverage TTA's generalizability while bypassing the huge computational overhead, we propose Self-TPT, an efficient Test-time Prompt Tuning method based on Self-supervised learning. Given the source data $\mathcal{S}$ and target data $\mathcal{T}$ with a potentially disjoint class set, our objective is to obtain prompts for VLMs to well-classify the images in $\mathcal{T}$. Self-TPT acquires task-specific knowledge from the source data and adapts these learned prompts to new classes at test time, without directly assessing the specific images from $\mathcal{T}$. The overall pipeline of Self-TPT, as depicted in Fig. 2, comprises three stages: prompt learning, test-time adaptation, and direct prediction. In `Stage 1`, we co-train the self-supervised task and the classification task:

$$\min_{\boldsymbol{\theta_p}, \boldsymbol{\theta_h}} \mathcal{L}_{ce}\left(\mathcal{X}^{(s)}, \mathcal{Y}^{(s)}, \mathcal{M}^{(s)}; \mathbf{P}, f, g\right) + \mathcal{L}_{ssl}\left(\mathcal{Y}^{(s)}; \mathbf{P}, g, h\right), \tag{5}$$

where $h(\cdot)$ is a SSL projection head, and $\boldsymbol{\theta_h}$ denotes its parameters. In `Stage 2`, given the class set $\mathcal{Y}^{(t)}$ of $\mathcal{T}$, we adapt using a text-oriented SSL task, decoupling the test-time adaptation from specific test samples $\mathcal{X}_i^{(t)}$:

$$\min_{\boldsymbol{\theta_p}} \mathcal{L}_{ssl}\left(\mathcal{Y}^{(t)}; \mathbf{P}, g, h\right). \tag{6}$$

The prompts refined through Eq. 6 are directly applied to predict samples in $\mathcal{T}$ without further adjustments, thereby streamlining the test-time adaptation into a pre-processing step and significantly

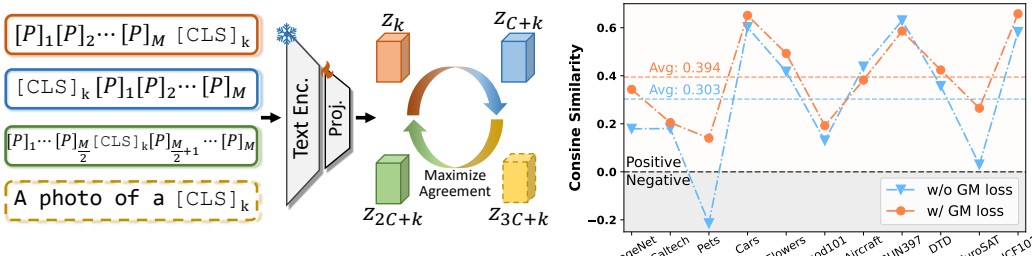

(a) **Contrastive Prompt Tuning**. Class token "`[CLS]`" is inserted at the end, front, and middle of the prompt sequences to generate positive pairs for contrastive learning. An additional hand-made prompt is incorporated as regularization.

(b) **Cosine similarity** between CPT and the classification task gradients across 11 datasets. A positive correlation can be observed between the two tasks. This correlation can be further enhanced by the gradient matching (GM) loss.

Figure 3: **Contrastive prompt tuning and gradient similarity analysis**.

reducing computational costs. In Sec. 3.3, we will detail the specific SSL task used in our Self-TPT framework, and in Sec. 3.4, we will empirically demonstrate how this SSL task facilitates improved classification performance.

## 3.3 CONTRASTIVE PROMPT TUNING

Within the Self-TPT framework, the core design is its self-supervised component. As indicated by Sun et al. (2020), the auxiliary task (in this case, the SSL task) should have a strong correlation with the main task (in this case, the classification task) to maintain effectiveness. In the realm of self-supervised learning, contrastive learning methods (Chen et al., 2020; He et al., 2020) have yielded impressive classification outcomes, even through the linear probing of frozen visual representations. This underscores a positive connection between contrastive tasks and classification tasks, thereby motivating our prioritization of contrastive learning in the SSL task design.

To construct the contrastive task, we adhere to the nature of classification that embeddings from the same class should align closely, while those from different classes should remain distinct to ensure inter-class distinguishability. To this end, we introduce a novel task named Contrastive Prompt Tuning (CPT). This task varies the insertion points of the class token "[CLS]" within prompt sequences to generate positive pairs. As depicted in Fig. 3a, CPT strategically places the "[CLS]" token at the end, front, and middle of the learnable prompts:

$$
\begin{aligned}
\boldsymbol{t}_k &= [P]_1[P]_2 \ldots [P]_M[\texttt{CLS}]_k, \\
\boldsymbol{t}_{C+k} &= [\texttt{CLS}]_k[P]_1[P]_2 \ldots [P]_M, \qquad\qquad k = 1, \ldots, C \\
\boldsymbol{t}_{2C+k} &= [P]_1 \ldots [P]_{\frac{M}{2}}[\texttt{CLS}]_k[P]_{\frac{M}{2}+1} \ldots [P]_M.
\end{aligned}
\tag{7}
$$

Furthermore, Zhu et al. (2022) suggests that a simple, hand-crafted prompt (*e.g.*, "`a photo of a`") embodies the general knowledge acquired during pre-training and can mitigate overfitting. With this insight, we incorporate such prompt into CPT to regulate the contrastive learning process:

$$
\boldsymbol{t}_{3C+k} = [\texttt{a}][\texttt{photo}][\texttt{of}][\texttt{a}][\texttt{CLS}]_k. \quad k = 1, \ldots, C
\tag{8}
$$

Eq. 7 and Eq. 8 establish four distinct views for each class. Let $i \in \{1, \ldots, 4C\}$ denote the index of a specific view from any class. We define the index set of its three other views as $\mathcal{P}_{(i)} \equiv \{i \bmod C, i \bmod C + C, i \bmod C + 2C, i \bmod C + 3C\} \setminus \{i\}$. The CPT loss is then formulated as:

$$
\mathcal{L}_{CPT} = -\sum_{i=1}^{4C} \log \frac{\sum_{j \in \mathcal{P}_{(i)}} \exp\left(\boldsymbol{z}_i \cdot \boldsymbol{z}_j / \tau\right)}{\sum_{j=1, j \neq i}^{4C} \exp\left(\boldsymbol{z}_i \cdot \boldsymbol{z}_j / \tau\right)}.
\tag{9}
$$

Here, $\boldsymbol{z}$ denotes the text feature after projection, computed as $\boldsymbol{z}_i = h(g(\boldsymbol{t}_i))$, and $\tau$ is a scaling temperature parameter, defaulting to 0.07.

Table 1: **Comparing inference computational costs.** "FPS" denotes frames per second. All tests are conducted on the same single A100 GPU and performed on ImageNet with a batch size of 1. Self-TPT-v is introduced in Implementation Details.

| Method | Venue | CLIP-B/16 | | CLIP-L/14 | |
|---|---|---|---|---|---|
| | | FPS ($\uparrow$) | Memory ($\downarrow$) | FPS ($\uparrow$) | Memory ($\downarrow$) |
| TPT (Shu et al., 2022) | NeurIPS'22 | 7.1 | 5.2GB | 3.3 | 8.5GB |
| DiffTPT (Feng et al., 2023) | ICCV'23 | 2.2 | 5.2GB | 1.0 | 8.5GB |
| PromptAlign (Samadh et al., 2023) | NeurIPS'23 | 5.3 | 11.2GB | 2.1 | 31.5GB |
| Self-TPT | – | **146.7** | **0.32GB** | **81.6** | **0.86GB** |
| Self-TPT-v | – | 29.3 | 0.66GB | 7.9 | 1.41GB |

## 3.4 GRADIENT MATCHING

A pertinent question arises: *how does CPT benefit classification during adaptation*? To uncover the underlying reasons, we conducted an empirical analysis by measuring the cosine similarity of gradients between the classification loss and the CPT loss during optimization, expressed as:

$$\cos(\nabla_{\mathcal{L}_{ce}}, \nabla_{\mathcal{L}_{CPT}}), \tag{10}$$

where $\nabla$ denotes the gradients corresponding to each loss. $\nabla\mathcal{L}_{ce}$ is computed as the average over all images in the dataset to ensure statistical significance. The resulting similarity scores, depicted as blue triangles in Fig. 3b, show positive correlations in 10 out of the 11 datasets examined. The direction of these gradients suggests that CPT can effectively align with the optimization pathways of the classification task during test-time adaptation, even in the absence of ground-truth labels.

Inspired by this finding, we propose a Gradient Matching (GM) loss to explicitly improve gradient similarity between the two tasks. During training, we noted that $\nabla_{\mathcal{L}_{ce}}$ exhibits sensitivity to small batch sizes. To obtain the stable optimization direction of classification, we employ exponential moving average (EMA) (Heckert & Filliben, 2003) to integrate both past and current gradient trends:

$$\widetilde{\nabla}_{\mathcal{L}_{ce}} = \alpha^T \nabla^0_{\mathcal{L}_{ce}} + \alpha^{T-1}(1-\alpha)\nabla^1_{\mathcal{L}_{ce}} + \ldots + (1-\alpha)\nabla^T_{\mathcal{L}_{ce}}. \tag{11}$$

Here, $\alpha$ is the decay rate, $\nabla\mathcal{L}ce^t$ denotes the gradient of the cross-entropy loss at step $t$, and $T$ represents the current step index. The GM loss is subsequently calculated as:

$$\mathcal{L}_{GM} = 1 - \cos(\widetilde{\nabla}_{\mathcal{L}_{ce}}, \nabla_{\mathcal{L}_{CPT}}). \tag{12}$$

Employing the GM loss, as observed in orange dots in Fig. 3b, yields notable increases in gradient similarity across 8 of the 11 datasets, indicating a strengthened correlation between the two tasks. In Sec. 4.3, we will demonstrate how this increase in similarity benefits downstream performance.

## 4 EXPERIMENTS

### 4.1 EXPERIMENTAL SETUP

**Datasets.** We use 11 datasets covering a diverse set of recognition tasks: ImageNet (Deng et al., 2009) and Caltech101 (Fei-Fei et al., 2004) for generic object recognition, OxfordPets (Parkhi et al., 2012), StanfordCars (Krause et al., 2013), OxfordFlowers (Nilsback & Zisserman, 2008), Food101 (Bossard et al., 2014) and FGVCAircraft (Maji et al., 2013) for fine-grained classification, SUN397 (Xiao et al., 2010) for scene recognition, DTD (Cimpoi et al., 2014) for texture classification, EuroSAT (Helber et al., 2019) for satellite recognition, and UCF101 (Soomro et al., 2012) for action recognition. Besides, four ImageNet variant datasets are involved: ImageNetV2 (Recht et al., 2019), ImageNet-Sketch (Wang et al., 2019), ImageNet-A (Hendrycks et al., 2021b) and ImageNetR (Hendrycks et al., 2021a).

**Evaluation settings.** We use three benchmarks for evaluation and report the top-1 accuracy: **(i)** Cross-data generalization: ImageNet is used as the source dataset, while the remaining 10 datasets serve as target datasets. **(ii)** Base-to-new generalization: Each dataset is divided equally into two subsets, base classes and new classes. The base subset is used as the source dataset, and the new subset is used as the target dataset for evaluation. **(iii)** Domain generalization: ImageNet is used as the source dataset, and 4 variant datasets are used as target datasets.

Table 2: **Cross-dataset generalization.** 16-shot ImageNet is used as the source dataset. We report top-1 accuracy (%) for each method across 10 target datasets. The best and second-best performances are highlighted in **bold red** and blue underline, respectively.

| Method | Caltech | Pets | Cars | Flowers | Food101 | Aircraft | SUN397 | DTD | EuroSAT | UCF101 | Avg. |
|---|---|---|---|---|---|---|---|---|---|---|---|
| CLIP (Radford et al., 2021) | 93.35 | 88.25 | 65.48 | 67.44 | 83.65 | 23.67 | 62.59 | 44.27 | 42.01 | 65.13 | 63.58 |
| *Prompt learning methods* | | | | | | | | | | | |
| CoOp (Zhou et al., 2022b) | 93.70 | 89.14 | 64.51 | 68.71 | 85.30 | 18.47 | 64.15 | 41.92 | 46.39 | 66.55 | 63.88 |
| CoCoOp (Zhou et al., 2022a) | 94.43 | 90.14 | 65.32 | 71.88 | 86.06 | 22.94 | 67.36 | 45.73 | 45.37 | 68.21 | 65.74 |
| KgCoOp (Yao et al., 2023) | 93.92 | 89.83 | 65.41 | 70.01 | 86.36 | 22.51 | 66.16 | 46.35 | 46.04 | 68.50 | 65.51 |
| LASP (Bulat & Tzimiropoulos, 2023) | 94.50 | 89.36 | 66.20 | 71.74 | 86.40 | 23.03 | 67.00 | 45.54 | 48.50 | 68.24 | 66.05 |
| MaPLe (Khattak et al., 2023a) | 93.53 | 90.49 | 65.57 | 72.23 | 86.20 | 24.74 | 67.01 | 46.49 | 48.06 | 68.69 | 66.30 |
| PromptSRC (Khattak et al., 2023b) | 93.60 | 90.25 | 65.70 | 70.25 | 86.15 | 23.90 | 67.10 | 46.87 | 45.50 | 68.75 | 65.81 |
| *LLM based methods* | | | | | | | | | | | |
| VisDesc (Menon & Vondrick, 2022) | 94.60 | 88.85 | 64.08 | 70.85 | 85.05 | 24.30 | 67.99 | 44.98 | 54.84 | 67.12 | 66.27 |
| WaffleCLIP (Roth et al., 2023) | 94.02 | 89.95 | 63.57 | 72.35 | 86.68 | 25.39 | 67.23 | 45.21 | **55.07** | 67.19 | 66.67 |
| *Test-time adaptation methods* | | | | | | | | | | | |
| TPT (Shu et al., 2022) | 94.16 | 87.79 | 66.87 | 68.98 | 84.67 | 24.78 | 65.50 | 47.75 | 42.44 | 68.04 | 65.10 |
| CoOp+TPT | 93.75 | 88.93 | 67.06 | 68.25 | 83.82 | 25.89 | 66.40 | 47.15 | 48.78 | 66.53 | 65.66 |
| MaPLe (Zhou et al., 2022b)+TPT | 93.59 | 90.72 | 66.50 | 72.37 | 86.64 | 24.70 | 67.54 | 45.87 | 47.80 | 69.19 | 66.50 |
| DiffTPT (Feng et al., 2023) | 92.49 | 88.22 | 67.01 | 70.10 | **87.23** | 25.60 | 65.74 | 47.00 | 43.13 | 68.22 | 65.47 |
| PromptAlign (Samadh et al., 2023) | 94.01 | 90.76 | 68.50 | 72.39 | 86.65 | 24.80 | 67.54 | 47.24 | 47.86 | 69.47 | 66.92 |
| Self-TPT | 94.09 | **91.83** | 66.66 | **72.60** | 86.89 | 25.41 | 67.75 | 49.02 | 52.94 | **70.05** | 67.72 |
| Self-TPT-v | **94.71** | 91.26 | **68.81** | 71.79 | 85.41 | **27.57** | **68.18** | **49.35** | 51.91 | 69.50 | **67.85** |

Table 3: **Base-to-new generalization.** The 16-shot base subset of each dataset is used as the source dataset. We report top-1 accuracy (%) on each new subset to evaluate the model's zero-shot performance to unseen classes. †: CoOp is reproduced under an identical experimental setup as ours.

| | Generic | | Fine-Grained | | | | | Specialized | | | | |
|---|---|---|---|---|---|---|---|---|---|---|---|---|
| Method | ImageNet | Caltech | Pets | Cars | Flowers | Food101 | Aircraft | SUN397 | DTD | EuroSAT | UCF101 | Avg. |
| CLIP (Radford et al., 2021) | 68.14 | 94.00 | 97.26 | 74.89 | 77.80 | 91.22 | 36.29 | 75.35 | 59.90 | 64.05 | 77.50 | 74.22 |
| *Prompt learning methods* | | | | | | | | | | | | |
| CoOp† (Zhou et al., 2022b) | 70.32 | 94.10 | 97.88 | 73.29 | 72.34 | 91.69 | 33.65 | 75.77 | 54.59 | 65.26 | 74.78 | 73.06 |
| CoCoOp (Zhou et al., 2022a) | 70.43 | 93.81 | 97.69 | 73.59 | 71.75 | 91.29 | 23.71 | 76.86 | 56.00 | 60.04 | 73.45 | 71.69 |
| ProDA (Lu et al., 2022) | 70.23 | 93.23 | 97.76 | 71.20 | 68.68 | 88.57 | 34.13 | 76.93 | 56.48 | 66.00 | 71.97 | 72.30 |
| KgCoOp (Yao et al., 2023) | 69.96 | 94.39 | 97.76 | 75.04 | 74.73 | 91.70 | 33.55 | 76.53 | 54.99 | 64.34 | 76.67 | 73.61 |
| LoGoPrompt (Shi & Yang, 2023) | 70.83 | 93.78 | 96.32 | 72.39 | 76.52 | 91.41 | 34.67 | 78.12 | 60.14 | 69.44 | 73.07 | 74.24 |
| LASP (Bulat & Tzimiropoulos, 2023) | 70.95 | 94.24 | **97.77** | 71.60 | 74.00 | 91.70 | 30.57 | 78.60 | 58.60 | **77.78** | 78.03 | 74.91 |
| RPO (Lee et al., 2023) | 71.57 | 94.37 | 97.50 | 75.53 | 76.67 | 90.83 | 34.20 | 77.80 | 62.13 | 68.97 | 75.43 | 75.00 |
| CoOp+SHIP (Wang et al., 2023c) | 69.95 | 95.20 | 97.87 | 73.90 | 74.40 | 91.03 | 32.33 | 75.27 | 56.88 | 66.87 | 76.85 | 73.69 |
| MaPLe (Khattak et al., 2023a) | 70.54 | 94.36 | 97.76 | 74.00 | 72.46 | 92.05 | 35.61 | 78.70 | 59.18 | 73.23 | 78.66 | 75.14 |
| PromptSRC (Khattak et al., 2023b) | 70.73 | 94.03 | 97.30 | 74.97 | 76.50 | 91.53 | 37.87 | 78.47 | 62.97 | 73.90 | 78.80 | 76.10 |
| *LLM based methods* | | | | | | | | | | | | |
| VisDesc (Menon & Vondrick, 2022) | 69.84 | 94.54 | 96.53 | 74.45 | 77.52 | 90.49 | 34.55 | 78.48 | 57.97 | 72.44 | 75.34 | 74.74 |
| WaffleCLIP (Roth et al., 2023) | 70.36 | 94.31 | 97.32 | 73.39 | **78.87** | 91.77 | 36.13 | 78.03 | 59.04 | 73.38 | 75.73 | 75.30 |
| *Test-time adaptation methods* | | | | | | | | | | | | |
| TPT (Shu et al., 2022) | 70.78 | 94.65 | 96.31 | 75.39 | 77.73 | 91.17 | 34.73 | 77.58 | 63.04 | 65.82 | 76.91 | 74.92 |
| CoOp+TPT | 72.58 | 94.87 | 97.65 | 75.15 | 72.34 | 91.73 | 36.95 | 77.05 | 58.82 | 64.90 | 69.44 | 73.77 |
| MaPLe+TPT | 72.24 | 94.29 | 97.37 | 75.20 | 72.10 | 92.03 | 35.81 | 79.18 | 59.91 | 68.96 | 77.34 | 74.95 |
| PromptSRC+TPT | 72.49 | 93.78 | 97.43 | 77.86 | 76.45 | 92.07 | 38.32 | 79.43 | 62.08 | 70.44 | 78.48 | 76.26 |
| PromptAlign (Samadh et al., 2023) | 72.59 | 94.50 | 97.56 | 75.71 | 72.34 | **92.68** | 37.27 | 79.48 | 60.55 | 72.71 | 79.30 | 75.88 |
| Self-TPT | 71.20 | 95.20 | **97.93** | 75.89 | 78.32 | 92.09 | 36.81 | 79.41 | 63.81 | 75.55 | **80.87** | 77.01 |
| Self-TPT-v | **73.40** | **95.49** | 97.15 | **78.08** | 77.99 | 91.46 | **38.33** | **79.92** | **65.14** | 74.74 | 80.49 | **77.47** |

**Implementation details.** Self-TPT is built on CoOp (Zhou et al., 2022b) and is implemented on CLIP-B/16 for evaluation. All results are averaged over three seeds. We set the number of prompts $M$ to 4. In stage 1, we use SGD as the optimizer with a learning rate of 0.002 and a batch size of 4. The prompts are trained on the source dataset in 16-shot manner. For the base-to-new setting, we train prompts for 10 epochs. For the cross-dataset and domain generalization setting, prompts are trained for 5 epochs. In stage 2, we update the prompts for 10 steps, using SGD as the optimizer with a learning rate of 0.1. Existing TPT-based methods utilize the input image and its 63 augmented views as input. To ensure a fair comparison, we also incorporate the 63 augmented images and perform an output ensemble. We refer to this specific approach as **Self-TPT-v**.

## 4.2 COMPARISON WITH THE STATE-OF-THE-ART METHODS

We compare Self-TPT with three kinds of methods: **i)** prompt learning methods that optimize prompts on a source dataset, **ii)** test-time prompt learning methods that adjust prompts at test time, **iii)** methods that leverage LLM (*e.g.*, GPT-3 (Brown et al., 2020)) to produce class descriptions.

**Computational costs.** Tab. 1 presents a comparison of the inference cost between Self-TPT and other TPT-based methods. Self-TPT achieves inference speeds 25 times faster than PromptAlign and 65 times faster than DiffTPT on CLIP-B/16, along with a significant reduction in graphics

Table 5: **Ablation study of Self-TPT.** "G.", "F.", and "S." represent "Generic", "Fine-Grained", and "Specialized" datasets, respectively. The default setting is colored grey.

(a) **Main components analysis.**

| CPT | TTA | GM | G. | F. | S. |
|-----|-----|-----|------|------|------|
| | | | 81.8 | 73.8 | 67.6 |
| ✓ | | | 82.3 | 74.9 | 70.5 |
| ✓ | ✓ | | 82.9 | 75.8 | 73.3 |
| ✓ | ✓ | ✓ | **83.2** | **76.2** | **74.9** |

(b) **Augmented views in CPT.**

| Views | G. | F. | S. |
|-------|------|------|------|
| All four views | **83.2** | **76.2** | **74.9** |
| w/o front | 83.0 | 75.8 | 74.4 |
| w/o mid | 83.1 | 76.1 | 73.9 |
| w/o hand | 83.1 | 75.9 | 73.7 |
| w/o hand-craft | 82.9 | 75.9 | 73.6 |

(c) **Study on Gradient Matching.**

| EMA | Matching | G. | F. | S. |
|-----|----------|------|------|------|
| | | 83.1 | 76.1 | 73.4 |
| | MSE | 82.8 | 75.6 | 72.2 |
| ✓ | MSE | 83.1 | 76.1 | 73.9 |
| | Cosine | 82.9 | 76.1 | 73.9 |
| ✓ | Cosine | **83.2** | **76.2** | **74.9** |

memory usage. Despite the use of an additional 63 augmented images, Self-TPT-v maintains a five-fold speed advantage and reduces memory usage by 15 times relative to PromptAlign. Given the rapid expansion of foundation models, the efficiency of Self-TPT highlights its potential for scalable stability in larger VLMs.

**Cross-dataset generalization.** In Tab. 2, we comapre the performance of Self-TPT with existing state-of-the-art methods in the cross-dataset setting. Our method outperforms the previously best method on 8 out of 10 datasets, yielding an average improvement of 0.93% over PromptAlign. Note that simply combining prompt learning with test-time adaptation doesn't always yield optimal outcomes. For instance, MaPLe+TPT shows only a slight improvement of 0.2% over MaPLe alone, suggesting that TPT may not consistently deliver significant performance improvements. Conversely, despite not being exposed to specific test images, Self-TPT still demonstrates superior performance, highlighting the effectiveness of our proposed framework.

**Base-to-new generalization.** In the base-to-new setting (Tab. 3), our method consistently outperforms others on 9 out of 11 datasets, achieving an average improvement of 1.37% over the previous best method, PromptSRC. Interestingly, while CoOp+TPT records a 0.71% improvement over CoOp, MaPLe+TPT shows a decline of 0.19%, again highlighting the potential unstable performance gains of TPT. Moreover, LLMs-based methods tend to fall short in complex scenarios requiring high-level understanding, *e.g.*, UCF101, which demands intricate human action comprehension.

**Domain generalization.** We present Self-TPT's results on four ImageNet variant datasets with domain shifts in Tab. 4. Remarkably, Self-TPT set new state-of-the-art records on three of these datasets, demonstrating its robustness to domain shifts and adaptability to varying image distributions. Although Self-TPT was outperformed by CoOp+TPT and CoOp+DiffTPT on ImageNet-V2, we speculate this is because ImageNet-V2's data distribution closely aligns with that of ImageNet, the source dataset. CoOp is well-known for overfitting on source data.

Table 4: **Domain generalization.** ImageNet is used as source data. "Aug" indicates the original image is augmented 63 times and input jointly.

| Method | Aug | IN-V2 | IN-Sk. | IN-A | IN-R | *Avg.* |
|--------|-----|-------|--------|------|------|------|
| CLIP | | 60.86 | 46.09 | 47.87 | 73.98 | 57.20 |
| TPT | ✓ | 63.45 | 47.94 | 54.77 | 77.06 | 60.81 |
| CoOp+TPT | ✓ | **66.83** | 49.29 | 57.95 | 77.27 | 62.84 |
| MaPLe+TPT | ✓ | 64.87 | 48.16 | 58.08 | 78.12 | 62.31 |
| DiffTPT | ✓ | 65.10 | 46.80 | 55.68 | 75.00 | 60.65 |
| CoOp+DiffTPT | ✓ | 66.80 | 49.50 | 58.09 | 73.90 | 62.07 |
| PromptAlign | ✓ | 65.29 | 50.23 | 59.37 | 79.33 | 63.56 |
| Self-TPT-v | ✓ | 66.55 | **51.72** | **63.48** | **79.76** | **65.38** |

## 4.3 ABLATION STUDY

**Main components analysis.** In developing Self-TPT based on CoOp, we made three key progress. First, we integrated CPT during the source prompt learning phase. This integration aims to cultivate more robust and generalizable feature representations. Second, we applied CPT during test-time adaptation, improving the understanding of the new class prior to the prediction phase. Lastly, we incorporated the GM loss in the source prompt learning stage to explicitly strengthen the gradient correlation between CPT and the classification task. The effectiveness of each component is quantitatively assessed in Tab. 5a. The results demonstrate that each component of Self-TPT contributes to consistent performance improvements across the board.

**Study on different views in CPT.** As shown in Fig. 3a, Self-TPT employs four distinct views per class for contrastive learning. In Tab. 5b, we conduct an ablation study by sequentially removing one view at a time. This results in a consistent decline in performance. Notably, removing the handcrafted view causes the most significant drop, as adapting the CLIP text branch independently without regularization may lead to misalignment with the visual branch. These findings suggest that incorporating multiple views enhances the effectiveness of the contrastive prompt tuning task.

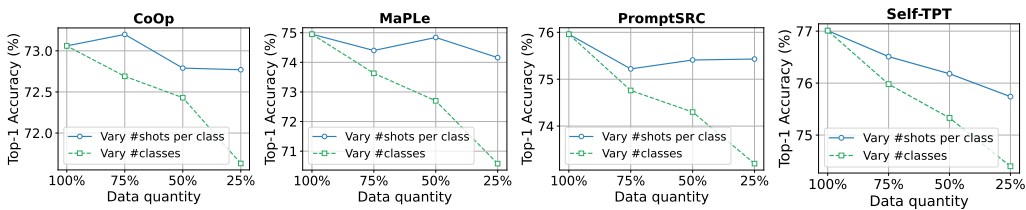

Figure 4: **Study on source data quality:** Is it better to have more classes or more shots?

Table 6: **Study on model versatility.**

(a) Self-TPT generalizes to different scales and architectures.

| Model | ResNet | | VisionTransformer | | |
|---|---|---|---|---|---|
| | RN50 | RN101 | B/32 | B/16 | L/14 |
| CLIP (Radford et al., 2021) | 68.72 | 69.90 | 71.80 | 74.22 | 80.34 |
| CoOp (Zhou et al., 2022b) | 66.75 | 68.98 | 70.26 | 73.06 | 78.97 |
| WaffleCLIP (Roth et al., 2023) | 69.04 | 70.06 | 72.30 | 75.30 | 81.12 |
| Self-TPT | **70.90** | **71.94** | **73.83** | **77.01** | **82.13** |

(b) Self-TPT generalizes to a different VLM.

| Method | Acc. |
|---|---|
| EVA-CLIP (Sun et al., 2023) | 77.33 |
| EVA-CLIP + CoOp (Zhou et al., 2022b) | 75.68 |
| EVA-CLIP + PromptSRC (Khattak et al., 2023b) | 78.68 |
| EVA-CLIP + VisDesc (Menon & Vondrick, 2022) | 78.12 |
| EVA-CLIP + WaffleCLIP (Roth et al., 2023) | 78.59 |
| EVA-CLIP + Self-TPT | **79.81** |

**Study on gradient matching.** Tab. 5c presents the results of the ablation study on the gradient matching (GM) loss. We replaced the cosine similarity loss with mean square error (MSE) in Eq. 12 and observed a performance decrease. This indicates that enforcing exact numerical equality of gradients from two different tasks may not be suitable. Additionally, we assessed the impact of using EMA and found consistent improvements, underscoring that maintaining a robust gradient direction is critical for the effectiveness of GM loss.

**Study on source data quantity and quality.** Source data is pivotal in both prompt learning and TPT-based methods. In Tab. 7, we examine the impact of reducing data quantity on model performance. The analysis encompasses 25%, 50%, 75%, and 100% of the default data volume. Our findings indicate that Self-TPT maintains robust performance even with limited source data, highlighting its efficiency in data utilization. Furthermore, we in-

Table 7: **Model performance with reduced data quantity.**

| Method | 25% | 50% | 75% | 100% |
|---|---|---|---|---|
| CoOp | 71.63 | 72.43 | 72.69 | 73.06 |
| MaPLe | 70.58 | 72.70 | 73.63 | 74.95 |
| PromptSRC | 73.20 | 74.30 | 74.76 | 75.96 |
| Self-TPT | **74.40** | **75.33** | **75.98** | **77.01** |

vestigate a critical question: Is it more beneficial to have more classes or more instances per class? This inquiry involves varying both the number of shots per class and the number of classes. We present the performance of four methods under this variation in Fig. 4. The results reveal a significant performance decline with a reduced number of classes, underscoring the importance of prioritizing class diversity over per-class instance quantity in data collection for real-world applications.

**Study on model versatility.** In Table 6a, we adapt Self-TPT to various backbone architectures, including ResNet (He et al., 2016) and ViT (Dosovitskiy et al., 2020), across different scales. Notably, Self-TPT consistently delivers performance improvements across all tested backbones, demonstrating its robust effectiveness. Furthermore, we extend the application of Self-TPT and several competitive methods to another VLM, EVA-CLIP (Sun et al., 2023), as illustrated in Table 6b. Once again, Self-TPT demonstrates distinct advantages over competing methods. In the context of rapid iteration in foundational models today, we believe Self-TPT can be seamlessly integrated with updated models to enhance their performance on downstream tasks.

## 5 CONCLUSION

In this paper, we introduce Self-TPT, a novel framework for efficient test-time prompt tuning. Self-TPT addresses the inefficiencies in inference found in existing TPT-based methods by using text-oriented self-supervised learning for new class adaptation, transforming per-image adaptation into a preprocessing step. We also introduce a novel contrastive prompt tuning (CPT) task for self-supervision. Empirical results show that CPT has a positive gradient correlation with classification tasks, highlighting its effectiveness. Based on this finding, we propose a gradient-matching loss to further enhance this correlation. Extensive experiments confirm the efficiency, effectiveness, and versatility of our method.

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

## A ADDITIONAL DETAILS

**Datasets Details.** Tab. 8 presents an overview of the 14 datasets utilized in the main paper. In line with CoOp (Zhou et al., 2022b), the classes "BACKGROUND_Google" and "Faces easy" are excluded from the Caltech101 dataset. As for the video dataset UCF101, we use the middle frame of each video as the input.

Table 8: **Datasets statistics.**

| Dataset | Classes | Train | Validation | Test |
|---|---|---|---|---|
| ImageNet (Deng et al., 2009) | 1,000 | 1.28M | N/A | 50,000 |
| Caltech101 (Fei-Fei et al., 2004) | 100 | 4,128 | 1,649 | 2,465 |
| OxfordPets (Parkhi et al., 2012) | 37 | 2,944 | 736 | 3,669 |
| StanfordCars (Krause et al., 2013) | 196 | 6,509 | 1,635 | 8,041 |
| Flowers102 (Nilsback & Zisserman, 2008) | 102 | 4,093 | 1,633 | 2,463 |
| Food101 (Bossard et al., 2014) | 101 | 50,500 | 20,200 | 30,300 |
| FGVCAircraft (Maji et al., 2013) | 100 | 3,334 | 3,333 | 3,333 |
| SUN397 (Xiao et al., 2010) | 397 | 15,880 | 3,970 | 19,850 |
| DTD (Cimpoi et al., 2014) | 47 | 2,820 | 1,128 | 1,692 |
| EuroSAT (Helber et al., 2019) | 10 | 13,500 | 5,400 | 8,100 |
| UCF101 (Soomro et al., 2012) | 101 | 7,639 | 1,898 | 3,783 |
| ImageNetV2 (Recht et al., 2019) | 1,000 | N/A | N/A | 10,000 |
| ImageNet-Sketch (Wang et al., 2019) | 1,000 | N/A | N/A | 50,889 |
| ImageNet-A (Hendrycks et al., 2021b) | 200 | N/A | N/A | 7,500 |
| ImageNet-R (Hendrycks et al., 2021a) | 200 | N/A | N/A | 30,000 |

**Additional Implementation Details.** The learnable prompts are initialized with pre-trained CLIP word embeddings of "a photo of a" at the beginning of stage 1. To eliminate the need for an additional validation set, we opt to select the model at the last step of both stage 1 and stage 2. Consistent with prior research (Zhou et al., 2022b;a; Zhu et al., 2022; Khattak et al., 2023a), training at stage 1 includes techniques such as random resized cropping and flipping. We also implement a warm-up strategy where the learning rate is initially set at $1e-5$ for the first epoch, and then it follows a cosine annealing schedule starting from $2e-3$. For the projection head, we employ a nonlinear projection similar to (Chen et al., 2020; Khosla et al., 2020), which incorporates an additional hidden layer with ReLU activation. By default, the projection dimension is set to 128. We initialize the weights of the projection head using the Xavier (Glorot & Bengio, 2010) and set the bias to 0. All experiments are conducted on a single A100 GPU.

## B ADDITIONAL STUDIES

**Study on hyperparameter sensitivity.** In Fig. 5, we present a sensitivity analysis of Self-TPT by exploring variations in the number of update steps and learning rates during the test-time adaptation phase. Notably, Self-TPT maintained robust performance across various adaptation steps, with a preference for higher learning rates to optimize its effectiveness.

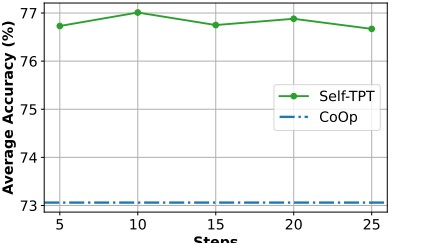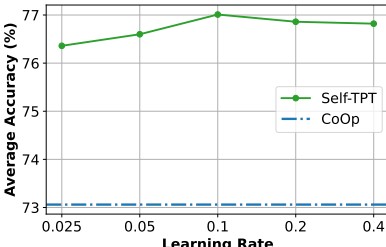

Figure 5: **Study on Hyperparameter Sensitivity.**

**Error bar analysis.** Error bar analysis was performed across both cross-dataset and base-to-new settings using three different seeds. The mean and standard deviation values are presented in Tab. 9 and Tab. 10. It was observed that, in most datasets, Self-TPT demonstrates a minimal performance divergence. Notably, EuroSAT, which has a limited number of classes, exhibited significant sensitivity in performance outcomes.

Table 9: **Error Bar** on cross-dataset generalization.

| Method | Caltech | Pets | Cars | Flowers | Food101 |
|---|---|---|---|---|---|
| CLIP | 93.35 | 88.25 | 65.48 | 67.44 | 83.65 |
| CoOp | 93.70 | 89.14 | 64.51 | 68.71 | 85.30 |
| **Self-TPT** | 94.09±0.17 | 91.83±0.26 | 66.66±0.38 | 72.60±0.29 | 86.89±0.10 |

| Method | Aircraft | SUN397 | DTD | EuroSAT | UCF101 | *Avg.* |
|---|---|---|---|---|---|---|
| CLIP | 23.67 | 62.59 | 44.27 | 42.01 | 65.13 | 63.58 |
| CoOp | 18.47 | 64.15 | 41.92 | 46.39 | 66.55 | 63.88 |
| **Self-TPT** | 25.41±0.45 | 67.75±0.04 | 49.02±0.29 | 52.94±2.42 | 70.05±0.06 | 67.73±0.18 |

Table 10: **Error Bar** on base-to-new generalization.

| Method | ImageNet | Caltech | Pets | Cars | Flowers | Food101 |
|---|---|---|---|---|---|---|
| CLIP | 68.14 | 94.00 | 97.26 | 74.89 | 77.80 | 91.22 |
| CoOp | 70.32 | 94.10 | 97.88 | 73.29 | 72.34 | 91.69 |
| **Self-TPT** | 71.20±0.02 | 95.20±0.18 | 97.93±0.20 | 75.89±0.62 | 78.32±0.44 | 92.09±0.20 |

| Method | Aircraft | SUN397 | DTD | EuroSAT | UCF101 | *Avg.* |
|---|---|---|---|---|---|---|
| CLIP | 36.29 | 75.35 | 59.90 | 64.05 | 77.50 | 74.22 |
| CoOp | 33.65 | 75.77 | 54.59 | 65.26 | 74.78 | 73.06 |
| **Self-TPT** | 36.81±0.14 | 79.41±0.28 | 63.81±0.41 | 75.55±0.82 | 80.87±0.42 | 77.01±0.11 |

**Visualization of CPT.** We employ t-SNE to visualize text embeddings and observe that embeddings generated using CPT demonstrate a distinct manifold structure, which is not apparent in the original embeddings. Additionally, an analysis of cosine distances between text embeddings indicates that CPT significantly enhances differentiation between classes.

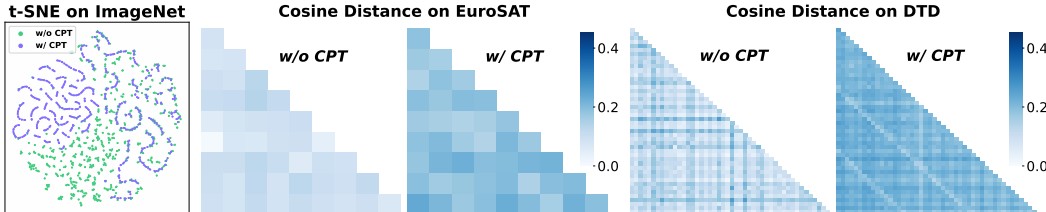

Figure 6: **Visualization of text embeddings.** We present a t-SNE visualization on the ImageNet dataset, consisting of 1000 classes, and compute the cosine distance between text embeddings from EuroAST and DTD. Darker colors indicate a greater distance.

**Extending Self-TPT to zero-shot video recognition** We assess the performance of Self-TPT in the realm of zero-shot video recognition. Following Open-VCLIP (Weng et al., 2023), we evaluate our model using the full UCF101 (Soomro et al., 2012) and HMDB51 (Kuehne et al., 2011) test sets, along with three selected subsets of Kinetics-600 (Carreira et al., 2018), as segmented by Chen & Huang (2021). Similar to Open-VCLIP, we incorporate neighbor-frame attention to model the temporal dynamics and train the CLIP image encoder on Kinetics-400 (Kay et al., 2017). It is important to note that the three subsets of Kinetics-600 have distinct class sets compared to Kinetics-400. For text prompts, we employ Self-TPT's prompts optimized on 16-shot ImageNet and then adapt them to the action classes at test time. The evaluation results are summarized in Tab. 11. Remarkably, Self-TPT surpasses the performance of the state-of-the-art Open-VCLIP by margins of 0.9%, 2.0%, and 0.7% across the respective datasets.

## C LIMITATIONS AND FUTURE WORK

Although existing TTA techniques show promise and effectiveness, their computational costs hinder their deployment in real-world scenarios. Self-TPT takes a stride forward by turning specific-sample adaptation into a pre-processing step. However, the extra costs brought by the nature of test-time adaptation still exist. There is a risk for Self-TPT degrading to vanilla TPT scenarios when the samples come with different class sets. In the future, our objectives include refining the Self-TPT framework to better align with practical applications. We also intend to investigate the potential of test-time prompt tuning across various other fields, such as more intricate visual-language tasks (Liu et al., 2024; Zhu et al., 2023).

Table 11: **Extending Self-TPT to zero-shot video recognition.**

| Method | UCF101 | HMDB51 | Kinetics-600 |
|---|---|---|---|
| CLIP (Radford et al., 2021) | 74.2 | 46.3 | 68.1±1.1 |
| ActionCLIP (Wang et al., 2021) | 77.4 | 48.0 | 62.5±1.2 |
| Text4Vis (Wu et al., 2023) | 76.4 | 44.5 | 60.1±0.5 |
| AdapterFormer (Chen et al., 2022c) | 80.5 | 50.5 | 67.0±0.4 |
| AIM (Yang et al., 2023) | 79.0 | 49.5 | 66.7±0.5 |
| ST-Adapter (Pan et al., 2022) | 77.9 | 50.3 | 60.2±1.8 |
| Open-VCLIP (Weng et al., 2023) | 83.5 | 53.2 | 73.0±0.8 |
| **Self-TPT** | **84.4** | **55.2** | **73.7±0.8** |

## D PSEUDO CODE

In Algo. 1, we present the pseudo-code for Self-TPT. The definitions of the symbols are consistent with those in the main paper.

---

**Algorithm 1** Self-TPT: Test-time Prompt Tuning with Self-supervision

---

**Input:** Source data $\mathcal{S} = \left(\mathcal{X}^{(s)}, \mathcal{Y}^{(s)}, \mathcal{M}^{(s)}\right)$, target data $\mathcal{T} = \left(\mathcal{X}^{(t)}, \mathcal{Y}^{(t)}\right)$
**Output:** Label index $I = (I_1, I_2, \dots)$
  # Stage 1: Prompt Learning
  **for** T = 1, 2, … **do**
    Sample batch $\left(\mathcal{X}_i^{(s)}, \mathcal{M}_i^{(s)}\right) \sim \left(\mathcal{X}^{(s)}, \mathcal{M}^{(s)}\right)$
    $\boldsymbol{t} \leftarrow \mathcal{Y}^{(s)}, \mathbf{P}$        # combine class token with prompts
    $\boldsymbol{e} \leftarrow f\left(\mathcal{X}_i^{(s)}\right), \boldsymbol{w} \leftarrow g\left(\boldsymbol{t}\right), \boldsymbol{z} \leftarrow h\left(\boldsymbol{w}\right)$        # compute image and text features
    $\mathcal{L}_{ce} \leftarrow \mathcal{L}_{ce}\left(\cos\left(\boldsymbol{e}, \boldsymbol{w}\right), \mathcal{M}_i^{(s)}\right)$        # supervised loss
    $\mathcal{L}_{CPT} \leftarrow \mathcal{L}_{CPT}\left(\boldsymbol{z}\right)$        # self-supervised loss
    $\nabla_{\mathcal{L}_{ce}^T} \leftarrow \partial\mathcal{L}_{ce}/\partial\boldsymbol{\theta_p}, \nabla_{\mathcal{L}_{CPT}} \leftarrow \partial\mathcal{L}_{CPT}/\partial\boldsymbol{\theta_p}$        # gradients w.r.t. prompts
    **if** T == 0 **then**
      $\widetilde{\nabla}_{\mathcal{L}_{ce}} \leftarrow \nabla_{\mathcal{L}_{ce}^T}$
    **else**
      $\widetilde{\nabla}_{\mathcal{L}_{ce}} \leftarrow \alpha\widetilde{\nabla}_{\mathcal{L}_{ce}} + (1-\alpha)\nabla_{\mathcal{L}_{ce}^T}$        # exponential moving average
    **end if**
    $\mathcal{L}_{GM} = 1 - \cos\left(\widetilde{\nabla}_{\mathcal{L}_{ce}}, \nabla_{\mathcal{L}_{CPT}}\right)$        # gradient matching loss
    $\boldsymbol{\theta} \leftarrow \boldsymbol{\theta} - \epsilon\left(\nabla_{\mathcal{L}_{ce}^T} + \nabla_{\mathcal{L}_{CPT}} + \nabla_{\mathcal{L}_{GM}}\right)$        # update model parameters
  **end for**
  # Stage 2: Test-time adaptation
  **for** T = 1, 2, … **do**
    $\boldsymbol{t} \leftarrow \mathcal{Y}^{(t)}, \mathbf{P}$        # combine class token with prompts
    $\boldsymbol{z} \leftarrow h\left(g\left(\boldsymbol{t}\right)\right)$        # compute text features
    $\mathcal{L}_{CPT} \leftarrow \mathcal{L}_{CPT}(\boldsymbol{z})$        # self-supervised loss
    $\boldsymbol{\theta_p} \leftarrow \boldsymbol{\theta_p} - \epsilon\left(\partial\mathcal{L}_{CPT}/\partial\boldsymbol{\theta_p}\right)$        # update prompt parameters
  **end for**
  # Stage 3: Make predictions
  $\boldsymbol{t} \leftarrow \mathcal{Y}^{(t)}, \mathbf{P}$        # combine class token with prompts
  $\boldsymbol{w} \leftarrow g\left(\boldsymbol{t}\right)$        # pre-compute text features
  **for** each $\mathcal{X}_i^{(t)} \sim \mathcal{X}^{(t)}$ **do**
    $\boldsymbol{e} \leftarrow f\left(\mathcal{X}_i^{(t)}\right)$        # compute image feature
    $I_i \leftarrow \arg\max \boldsymbol{e} \cdot \boldsymbol{w}$        # directly make prediction
  **end for**

---

