# OpenReview forum: "Efficient Test-Time Prompt Tuning for Vision-Language Models"
_ICLR.cc/2025/Conference — Submitted to ICLR 2025_

### Official Review · Reviewer_ZKen · 2024-10-27

**Soundness:** 2
**Presentation:** 4
**Contribution:** 2
**Rating:** 3
**Confidence:** 4

**Summary:**

While previous test-time prompt tuning methods have demonstrated their effectiveness, they are computationally expensive as they require multiple forward and backward passes on each test sample. In this paper, the authors propose an efficient test-time prompt tuning technique to mitigate this overhead through self-supervised learning on predefined class names. To enhance the alignment between the classification task and the self-supervised task, they further introduce a gradient matching loss. On base-to-new, cross-dataset, and domain generalization benchmarks, the proposed method achieves superior efficiency and performance compared to state-of-the-art approaches.

**Strengths:**

- The paper is well-written and easy to follow.
- The proposed method is simple and straightforward to implement.
- The motivation to reduce test-time overhead is meaningful. The method avoids multiple forward passes and real-time tuning, making it practical for implementation.

**Weaknesses:**

1.   The proposed method faces a fundamental technical flaws when adapting models to new target domains that share the same label space as the source domain. The unsupervised prompt learning framework operates solely on the text branch and is designed to adapt to new class names. However, it overlooks potential domain shifts in the image inputs. For example, if the source domain consists of real photos and the target domain consists of sketches with the same labels, the proposed method is unlikely to adapt effectively to the sketches without leveraging the information from the input images. This raises doubts about the results reported in Table 4. I conjecture the improvements may instead stem from prompt ensembling.

2.   The design of the ablation studies is problematic. It is well established that learning multiple soft prompts can outperform a single prompt [1,2]. From my experience with prompt tuning, using four prompts and properly ensembling them can improve accuracy by more than 1% (as also demonstrated in the G+E row of Table 2 in [1]). The ablation studies, however, do not include the performance of comparable baselines that use similar ensembling techniques.

[1] *PLOT: Prompt Learning with Optimal Transport for Vision-Language Models*
[2] *Prompt Distribution Learning*

**Questions:**

Please refer to the weakness.

---

> ### Author Response · Authors · 2024-11-24
>
> We sincerely appreciate your constructive feedback. Please allow us to address your concerns below.
>
> **W1 (part1): The proposed method faces a fundamental technical flaw when adapting models to new target domains that share the same label space as the source domain.**
>
> Thank you for your comment. We acknowledge that Self-TPT is **not specifically designed for domain adaptation**. However, we respectfully disagree that this aspect constitutes a fundamental technical flaw in our paper, for the following reasons:
>
> - **Purpose of our study:** This paper primarily investigates test-time **new class understanding** (as opposed to domain adaptation). We demonstrate that our proposed method achieves sota performance with high efficiency in this area by benchmarking it against two popular open-vocabulary settings.
> - **Orthogonality to domain adaptation methods:**  It is crucial to highlight that **Self-TPT can be seamlessly integrated with TPT-like methods**, which performs single-sample prompts adaptation. It offers the flexibility for users to choose between prioritizing computational efficiency (using Self-TPT alone) or enhanced domain adaptation performance (combining Self-TPT with TPT).
> - **Decent domain adaptation performance:** Although Self-TPT is not tailored for domain adaptation, it performs well in such benchmarks (shown in Table 4). This performance makes the term "a fundamental technical flaw" an unsuitable descriptor for the Self-TPT's limitations in domain adaptation contexts.
>
> **W1 (part2): This raises doubts about the results reported in Table 4. I conjecture the improvements may instead stem from prompt ensembling.**
>
> Thank you for your inquiry. To address your concern, we have compared our method with the two prompt ensembling techniques you referenced. The table below shows that although all methods incorporate prompt ensembling, our approach consistently outperforms others. This suggests a significant performance gap between Self-TPT and prompt ensembling techniques.
>
> | Method (CLIP-B/16) | # Ensembled prompts | Base-to-new | Cross-dataset | Domain generalization |
> | ------------------ | ------------------- | ----------- | ------------- | --------------------- |
> | ProDA              | 32                  | 72.30       | 65.62         | --                    |
> | PLOT++             | 4                   | --          | 64.19         | 58.38                 |
> | Self-TPT-v (ours)  | 4                   | 77.47       | 67.85         | 65.38                 |
>
> **W2: The ablation studies do not include the performance of comparable baselines that use similar ensembling techniques.**
>
> Thank you for your comment. **We have indeed included the experiment you mentioned** in our original submission. Please refer to Line 2 of Table 5(a) for the results.
>
> As stated in Lines 419-425 of our paper, Table 5(a) provides the following breakdown of our method: Line 1 shows results for our baseline CoOp, **Line 2 describes CoOp enhanced with CPT which incorporates four prompt perspectives and employs contrastive learning**. Line 3 further employs CPT at test time for new class adaptation, while Line 4 incorporates the gradient matching loss we proposed. Notably, Self-TPT surpasses the results of Line 2 by margins of 0.9%, 1.3%, and 4.4% on the "Generic," "Fine-Grained," and "Specialized" datasets, respectively.

---

> > ### Comment · Reviewer_ZKen · 2024-11-24
> >
> > Thank you for the authors’ responses. However, they do not address my concerns.
> >
> > (1) In domain generalization settings, the training and testing datasets share the same class names (no changes in class priors.). **Theoretically, this should negate any significant improvements achieved through your method**. Yet, improvements are still observed in your results. I find this inconsistency unconvincing.
> >
> > (2) The results in Table 5(a) **are not the experiment I referred to**.  My concern lies with the effectiveness of CPT. The experiment should involve learning 4 soft prompts independently using cross-entropy and then ensembling them.

---

### Official Review · Reviewer_phCX · 2024-11-01

**Soundness:** 3
**Presentation:** 2
**Contribution:** 2
**Rating:** 6
**Confidence:** 3

**Summary:**

This paper proposes Contrastive Prompt Tuning (CPT). CPT is a text-only self-supervised object to refine class embedding. This additional refinement improves standard supervised prompt learning. Since CPT is text-only, it can also be applied to target class names, without accessing the test images like Test-time Prompt Tuning (TPT). There is also a finding on gradient similarity of CPT and classification task, which motivates the gradient matching loss to further improve the performance.

**Strengths:**

- CPT during supervised prompt learning does complement the supervised objective.
- Since CPT is text-only, it can also be applied to target class names, without accessing test images. This avoids per-sample optimization, which is costly.
- CPT partially mimics supervised learning gradient, and motivated by this, the paper proposes a gradient matching loss to further enhance the performance.
- good experiments to support the findings in the paper

**Weaknesses:**

Since CPT does not use test images, it can also be treated just as a additional loss to supplement supervised prompt learning.
- might be good to compare with more recent prompt learning approaches (Any-shift prompting, PromptKD etc), and not just focus on test-time prompt tuning approaches, where there are less recent baselines.
- can CPT also be combined with TPT for each sample like other prompt tuning approaches? self-TPT-v just do output ensemble?

**Questions:**

See weaknesses

---

> ### Author Response · Authors · 2024-11-24
>
> We sincerely appreciate your insightful feedback. Please allow us to address your concerns below.
>
> **W1: Compare with more recent prompt learning approaches (Any-shift prompting, PromptKD etc).**
>
> Thank you for your suggestion. We will incorporate Any-shift prompting into Tables 2-4. Additionally, we will discuss PromptKD in our related work section, as a direct comparison with our method would not be appropriate due to its use of a larger pre-trained model for knowledge distillation. We will also include more recent prompt learning studies in accordance with your recommendations.
>
> **W2: Can CPT also be combined with TPT?**
>
> Yes, your observation is correct. Our method is orthogonal to existing test-time prompt tuning methods, such as TPT. Below are the results demonstrating their integration:
>
> | Method         | ImageNet | Caltech | Pets  | Cars  | Flowers | Food101 | Aircraft | SUN397 | DTD   | EuroSAT | UCF101 | Base2new avg. acc |
> | -------------- | -------- | ------- | ----- | ----- | ------- | ------- | -------- | ------ | ----- | ------- | ------ | ----------------- |
> | Self-TPT       | 71.20    | 95.20   | 97.93 | 75.89 | 78.32   | 92.09   | 36.81    | 79.41  | 63.81 | 75.55   | 80.87  | 77.01             |
> | Self-TPT + TPT | 72.82    | 95.12   | 97.67 | 77.24 | 78.96   | 92.57   | 37.77    | 80.28  | 65.98 | 76.86   | 81.68  | 77.91             |

---

> ### Comment · Reviewer_phCX · 2024-11-25
>
> Thank you for your response and experiments. I think CPT does compliment existing techniques. But I'm not sure whether it is best presented as an efficient test-time prompt tuning technique. I maintain the original score.

---

### Official Review · Reviewer_bs2R · 2024-11-03

**Soundness:** 2
**Presentation:** 3
**Contribution:** 3
**Rating:** 5
**Confidence:** 3

**Summary:**

This paper introduces Self-TPT, a novel framework for efficient test-time prompt tuning that addresses computational inefficiencies in existing methods. It proposes the Contrastive Prompt Learning (CPT) to minimize the intra-class distance, and a gradient matching loss to further enhance it.

**Strengths:**

1.	The approach is efficient to test-time adaptation that significantly reduces computational costs.
2.	The Comprehensive experiments across multiple datasets shows the effectiveness.

**Weaknesses:**

1.	The novelty is not significant, as leveraging multiple prompts is proved to be useful in Kgcoop and Promptalign paper, and the Gradient matching loss provide very little improvement and may loss performance in some settings, as shown in table 5c.
2.	Do you need more prompts than others, since you may need three more prompts for CPT.

**Questions:**

1.	Can it be applied with existing prompt learning methods, like CoOp + self-TPT
2.	In table 5c, what is the experiment setting for the first line? I assume it should has the same numbers as line 3 in table 5a, but they are different. Could you clarify that?

---

> ### Author Response · Authors · 2024-11-24
>
> We appreciate the time and effort you have dedicated to reviewing our paper. Please allow us to address your concerns below.
>
> **W1: The novelty is not significant, as leveraging multiple prompts is proved to be useful in Kgcoop and Promptalign paper. The Gradient matching loss provide very little improvement and may loss performance in some settings, as shown in table 5c.**
>
> Thank you for your comment.
>
> First, we would like to clarify that our primary novelty is not in using multiple prompts, but in **developing the first text-only, self-supervised test-time adaptation framework that effectively balances performance with efficiency**. Within this framework, we implemented CPT as a specific self-supervised task. Additionally, we studied the gradient correlation phenomenon to improve the understanding of our framework's effectiveness.
>
> Second, **our study fundamentally differs from prior works**. KgCoOp utilizes hand-crafted prompts for distillation and PromptAlign employs multi-layer and multi-modal prompts, while **our innovation lies in adjusting the class token positions for contrastive learning, which has not been proposed before.** Moreover, our method surpasses KgCoOp and PromptAlign in the base-to-new setting by 3.86% and 1.59% respectively, signifying the substantial progress we have made.
>
> Lastly, Table 5(c) details our **trial-and-error experiments**, thus, some configurations did lead to decreased performance. [1, 2] have shown that manipulating gradients to meet specific expectations is challenging. Our GM loss both improves the task gradient similarity and generally boosts performance. While the improvements in some scenarios are modest, we believe our results could motivate future exploration of test-time prompt tuning.
>
> **W2: Do you need more prompts than others, since you may need three more prompts for CPT.**
>
> Thank you for your inquiry. Our approach **does not require additional prompts**; we utilize the same number of prompts as our baseline, CoOp. Specifically, we employ four $1\times512$ vectors as prompts. Importantly, the class token is inserted at varying positions within the prompt sequences to create distinct views, while the prompts themselves are shared across these views.
>
> **Q1: Can it be applied with existing prompt learning methods?**
>
> Yes, Self-TPT can be integrated with existing prompt learning methods. Applying Self-TPT involves two steps: 1) conducting CPT during the training phase on source data and optionally adding $\mathcal{L}_{GM}$ to enhance performance; 2) applying CPT to new classes during test-time for adaptation prior to making predictions.
>
> **Q2:  In Table 5c, what is the experiment setting for the first line?**
>
> You are correct, and I apologize for any confusion. Line 1 of Table 5c and Line 3 of Table 5a have identical settings. However, these two experiments were conducted on different machines, which led to slight differences in the results. These variations are within the expected range, as detailed in our error bar analysis in Table 9.
>
> [1] Gradient Projection Memory for Continual Learning. ICLR 2021.
>
> [2] Fishr: Invariant Gradient Variances for Out-of-Distribution Generalization. ICML 2022.

---

> > ### Comment · Reviewer_bs2R · 2024-11-25
> >
> > Thanks for the authors’ responses. However, some concerns remain unaddressed:
> >
> > 1. Missing experiment on CoOp + self-TPT: I believe this should be included as a basic baseline since CoOp is a widely used prompt learning method. It would be helpful to present results demonstrating the effectiveness of their method.
> > 2. Limited improvement with GM loss: A deeper analysis of the inconsistency would enhance the understanding of its impact.
> >
> > Therefore, I would prefer to maintain my score.

---

### Official Review · Reviewer_dxNg · 2024-11-03

**Soundness:** 3
**Presentation:** 3
**Contribution:** 2
**Rating:** 6
**Confidence:** 4

**Summary:**

The authors proposed Self-TPT, an efficient framework for test-time prompt tuning. This framework integrates contrastive prompt tuning (CPT) during the source prompt learning phase to cultivate more robust and generalizable feature representations. It applies CPT during test-time adaptation to improve the understanding of new classes and introduces a gradient matching loss in the source prompt learning phase to enhance the gradient correlation between CPT and classification tasks. Evaluation on three challenging zero-shot benchmarks shows that Self-TPT significantly reduces inference costs while achieving state-of-the-art performance, effectively balancing the trade-off between efficiency and efficacy.

**Strengths:**

1.The authors constructed contrastive prompt learning to enhance class differences and further strengthened this correlation through gradient matching. The idea of varying the position of the class token for text augmentation is novel to me.

2.The authors conducted comprehensive comparative experiments on three challenging benchmark datasets using their proposed method. The results demonstrated the method's outstanding performance.

3.The proposed Self-TPT method has lower inference computational costs, highlighting its potential for scalable stability in larger visual language models.

**Weaknesses:**

1.The authors vary the insertion points of the class token within prompt sequences as data augmentation to create positive pairs. However, the authors fail to give a convincing argument as to why that should lead to a better performance. For example, how did the authors decide to position the class token at the front, middle, and end, rather than choosing the positions randomly?

2.It is hard to convince me that applying a strong constraint (the GM loss) on contrastive loss and the CE loss? Constrastive loss is a more robust one possibly with better generalization ability, while the CE loss has the absolutely right discriminative information. I cannot see obviously strong relation. Exp in Table 5c, cannot convince me. Better provide more evidence or theoretical analysis.

3.In the prompt learning process of Stage 1, how are the weights among Lce, LCPT, and LGM distributed? The authors need to provide more relevant experimental results. I would also suggest authors to compare their method with more recent  studies, there are many related studies in 2024.

4. There may be a writing error in Table 5, where 'w/o hand' should probably be 'w/o end' ?

**Questions:**

Refer to weakness,

---

> ### Author Response · Authors · 2024-11-24
>
> Thank you for your detailed review and insightful comments. Please allow us to address your concerns below.
>
> **W1 (part1): The authors vary the insertion points of the class token, and fail to give a convincing argument as to why that should lead to a better performance.**
>
> Thank you for your comment. Our CPT brings performance gain in both the source training and test-time adaptation stages, which we detail as follows:
>
> - **Source training:** We alter the insertion points to generate diverse views of a class. This is analogous to image data augmentations such as random rotation. Such augmentation forms the basis for contrastive learning. This not only aids in learning **more robust prompts with enhanced generalization ability** (as you mentioned in W2) but also **ensures inter-class distinguishability** (as demonstrated in Figure 6).
> - **Test-time adaptation:** [1] theoretically substantiates that **positive gradient correlation** between loss functions contributes to improved TTA performance. We discuss this in Section 3.4 and empirically demonstrate in Figure 3(b) that our proposed **CPT consistently exhibits a positive gradient correlation with the classification task** in 10 datasets.
>
> **W1 (part2): How did the authors decide to position the class token at the front, middle, and end, rather than choosing the positions randomly?**
>
> Thank you for your question. We aimed to ensure the simplicity of our method by making it straightforward to demonstrate and easy to replicate, so we chose to insert the class token at fixed positions. The random insertion is certainly feasible for Self-TPT, and we present the results below.
>
> | Augmentation    | Base2new acc. |
> | --------------- | ------------- |
> | 3 fixed points  | 77.01         |
> | 3 random points | 76.89         |
>
> **W2: Contrastive loss is a more robust one possibly with better generalization ability, while the CE loss has the absolutely right discriminative information. I cannot see obviously strong relation. Exp in Table 5c, cannot convince me.**
>
> Thank you for your comment. Self-supervised loss, to which contrastive loss belongs, can indeed display a strong relation with classification loss, as evidenced by studies [1, 2, 3]. We demonstrate the relation between CPT and CE loss by analyzing their backpropagated gradients across 11 datasets (Section 3.4 and Figure 3(b)). Additionally, Table 5(c) is not intended to illustrate the relationship between contrastive loss and CE loss; rather, it serves as a trial-and-error experiment to explore the effective design for gradient matching loss.
>
> **W3:  How are the weights among losses distributed? The authors need to provide more experiments. I would also suggest authors to compare their method with more recent studies.**
>
> Thank you for pointing this out. Currently, the weights for three losses are each set to 1. In response to your suggestion, we have conducted an ablation study to explore different weight distributions:
>
> | Weight of  $\mathcal{L}_{ce}$ | Weight of $\mathcal{L}_{CPT}$ | Weight of $\mathcal{L}_{GM}$ | Base2new acc. (before TTA) | Base2new acc.(after TTA) |
> | ----------------------------- | ----------------------------- | ---------------------------- | -------------------------- | ------------------------ |
> | 1                             | 1                             | 1                            | 74.66                      | 77.01                    |
> | 2                             | 1                             | 1                            | 73.51                      | 76.14                    |
> | 1                             | 2                             | 1                            | 74.29                      | 76.75                    |
> | 1                             | 1                             | 2                            | 74.01                      | 76.49                    |
>
> The results suggest that increasing the weight for CE loss exacerbates overfitting on base classes. Meanwhile, increasing weights for CPT or GM loss slightly hampers the learning of classification knowledge. Our default setting provides the best balance across the configurations tested.
>
> Additionally, we plan to compare our method with more contemporary studies as you suggested.
>
> **W4: There may be a writing error in Table 5.**
>
> Thank you for pointing this out. We apologize for this oversight and will ensure it is corrected in our final version.
>
> ---
> [1] Test-Time Training with Self-Supervision for Generalization under Distribution Shifts. ICML 2020.
>
> [2] TTT++: When Does Self-Supervised Test-Time Training Fail or Thrive? NeurIPS 2021.
>
> [3] Test-Time Training with Masked Autoencoders. NeurIPS 2022.

---

> > ### Comment · Reviewer_dxNg · 2024-11-25
> >
> > Thank you for your response and your extra experimental results. But I cannot be fully convinced regarding the marginal performance gain, and I still cannot buy the benefits of strong constraints between two losses. I will maintain my original score.

---

### Meta-Review · Area_Chair_dFUa · 2024-12-17

**Metareview:**

The paper aims to reduce the computational overhead of test-time prompt tuning methods by applying self-supervised learning to new class names instead of using test images for model update. The paper received four reviews with 3x borderline and 1x reject ratings. In general, the reviews are negative. The reviewers raised questions about the novelty and are not convinced by the current results as important experiments are missing. The reviewers also pointed out that the improvement of the proposed loss is not significant enough. The reviews suggest that the paper has many lingering questions that need to be addressed. Therefore, the paper is not ready for publication at ICLR.

**Additional Comments On Reviewer Discussion:**

The reviewers asked several questions about missing experiments and comparisons but the rebuttal failed to provide accurate results. The reviewers indicated that they were not satisfied with the rebuttal and thus did not increase the rating.

---

### Decision · Program_Chairs · 2025-01-22

Reject